# Physiotherapy Improves Dogs’ Quality of Life Measured with the Milan Pet Quality of Life Scale: Is Pain Involved?

**DOI:** 10.3390/vetsci9070335

**Published:** 2022-07-02

**Authors:** Patrizia Piotti, Mariangela Albertini, Elisa Lavesi, Annalisa Ferri, Federica Pirrone

**Affiliations:** 1Department of Veterinary Medicine and Animal Sciences, University of Milan, Via dell’Università 6, 26900 Lodi, Italy; patrizia.piotti1@unimi.it (P.P.); elisa.lavesi@gmail.com (E.L.); federica.pirrone@unimi.it (F.P.); 2Zampe Amore e Fisioterapia, Via Leopoldo Penagini, 9, 26838 Tavazzano con Villavesco, 26900 Lodi, Italy; vet.annalisa@gmail.com

**Keywords:** physiotherapy, physiatric exam, dog, pain, quality of life, well-being

## Abstract

**Simple Summary:**

Recent scientific evidence highlights the importance of assessing quality of life in veterinary patients. Quality of life reflects the well-being of animals from the physical, psychological, and social point of view, as well as the safety and freedom within their environment. In the current study, the quality of life of 20 adult dogs was measured at the beginning and at the end of a physiotherapy treatment, and the relationship with their clinical symptoms was investigated. The dogs underwent various physiotherapy procedures due to neurological, orthopedic, and degenerative conditions and had various degrees of lameness and pain. The assessment of the severity of the condition, diagnosed by the clinician, and the observations on the physical quality of life provided by the dog owners were aligned. In addition, the analysis indicated that the psychological quality of life of the dogs improved following the physiotherapy treatment, highlighting the emotional impact of the medical conditions on the dogs. Furthermore, the social quality of life was negatively impacted by the severity of the lameness, emphasizing how the medical conditions affected all aspects of the dog’s life. These results reveal the importance of considering psychological and emotional aspects when assessing the health of veterinary patients.

**Abstract:**

Quality of life is defined as an individual’s satisfaction with its physical and psychological health, its physical and social environment, and its ability to interact with the environment. Understanding companion dogs’ QOL can help veterinarians and owners know when treatment options have successfully alleviated symptoms of disease in such fields as veterinary physiotherapy. For this study, 20 adult dogs were selected from patients of a physiotherapy referral center with orthopedic, neurological, and/or degenerative conditions. The severity of the medical problem was ranked, and the symptoms, the treatment plan, and demographic data were recorded at the time of the physical examination. In addition, the owner of the dog was asked to fill out a questionnaire on the quality of life of the pet (the Milan Pet Quality of Life scale) at the time of the first consultation as well as the last follow-up after the treatment. The MPQL measures four domains of QOL: physical (signs of medical conditions), psychological (emotional and behavioral well-being), social (quality and extent of social interactions), and environmental (freedom and safety in one’s environment). The results of the study indicated a significant improvement in the psychological QOL domain following physiotherapeutic treatment. The social QOL domain declined with the severity of lameness, while the physical QOL, as reported by the owner, declined with the overall criticality of the medical condition, as ranked by the physiotherapist. The results of the study support the recent evidence of a relationship between pain and canine psychological well-being and highlight the importance of investigating psychological and emotional aspects of dogs’ QOL when treating orthopedic and neurological cases with physiotherapy.

## 1. Introduction

Quality of life (QOL) is reflective of an individual’s well-being, which can be evaluated through psychometric scales and observed in the behavior of pets [1]. The World Health Organization (WHO) states that QOL models should measure life experiences relevant to the individual, reflect the variety of life aspects, consider contexts beyond physical welfare, and measure experiences at the population and individual levels [2,3]. The WHO provided a tool to measure the four dimensions of QOL, i.e., physical health (daily activities, dependence on medicinal substances and aids, energy and fatigue levels, mobility, pain and discomfort, sleep and rest, working capacity), psychological health (bodily image and appearance, negative and positive feelings, self-esteem, spirituality, religion, personal belief, thinking, learning, memory and concentration), social well-being (personal relationships, social support, sexual activity), and environmental welfare (financial resources, freedom, physical safety and security, accessibility and quality of health and social care, home environment, opportunities for acquiring new information and skills, participation in and opportunities for recreation/leisure activities, quality of physical environment (pollution/noise/traffic/climate), transport) [4].

When looking at non-human animals, McMillan defined quality of life (QOL) as “a multidimensional balance between pleasure states and discomfort” [5]. Wojciechowska et al. [6] added that QOL “comprises the state of the animal’s body and mind and the extent to which its nature or telos (genetic traits manifested through breed and temperament) is satisfied”. Schneider et al. [7] provided further experimental evidence of the multidimensional nature of QOL measures in dogs. For the purpose of this work, QOL is defined, as indicated by Belshaw et al. [8], as “an individual’s satisfaction with its physical and psychological health, its physical and social environment, and its ability to interact with that environment”. In the veterinary field it is believed that understanding companion dogs’ QOL can help veterinarians and owners know when treatment options have successfully alleviated symptoms of disease, when further care is necessary, or when further care outweighs the benefits on the dogs’ QOL [7]. For example, it is important, when the animal may be suffering from painful conditions and reduced mobility, to consider the emotional state in welfare evaluations [9,10]. Veterinary physiotherapy is one field that would benefit from the use of QOL measures: the overarching goal of physiotherapeutic treatments is often the improvement of the overall QOL of the dog rather than the simple remedy of the physical damage or recovery of loss of function [11]. Physiotherapy, through personalized rehabilitative programs, improves the mobility, pain, and the physical challenges that come from injuries, disease, and other medical conditions. In the veterinary field, it is largely recognized as a fundamental clinical discipline. Specifically, veterinary physiotherapy involves the assessment and treatment of the musculoskeletal and neurological conditions [12]. Currently, the QOL of clinical patients is not routinely assessed in the veterinary field. However, this would be particularly beneficial for those patients, such as physiotherapy patients, with conditions that affect multiple aspects of their life, including mobility, social interactions, and general discomfort due to painful conditions.

Limitations of several canine QOL measures, however, are that (1) they are not optimized for online or self-reporting use by the dog owners [6], (2) they do not rely on unambiguous scales [7], and (3) they are limited to health-related QOL rather than being multidimensional [13]. Recently, however, the Milan Pet Quality of Life scale (MPQL) was developed as an owner-reported measure of the QOL of cats and dogs during the COVID-19 pandemic [14]. The MPQL summarizes four QOL domains: physical (with facets including medical conditions, mobility limitations due to pain, and play), psychological (with facets including signs of negative affect, anxiety symptoms, sleep, appetite, being subjected to uncomfortable medical or grooming procedures), social (with facets including training, time together, feeding time, and signs of positive interactions), and environmental (with facets regarding freedom and safety within the home and external environment) [14]. The scale has been partially validated for use in Italian and English and its measures have been assessed against measurements of a pet’s individual differences, the pet management, and the pet–owner relationship [14].

The aim of the current project was to use the MPQL to investigate the changes in the QOL of a group of privately owned family dogs, comparing their QOL at the beginning and at the end of a physiotherapy treatment. We predicted all four domains of QOL to improve, due to the improved affect and physical health following the reduction in pain, improved social interactions due to re-gained mobility, and increased freedom within their environment.

## 2. Materials and Methods

### 2.1. Subjects

Dogs were selected, through convenience sampling, from patients brought to the referral center “Zampe Amore e Fisioterapia” for a physiotherapy referral for orthopedic, neurological, and/or degenerative conditions (Appendix A). Overall, 20 adult dogs were included in the study (Mdn_age_ = 4.5 years, range 1–14 years, F = 8, neutered = 10). The dog owners gave their consent during the clinical veterinary consultation performed by the physiotherapist (A.F.), who examined the dogs according to standard veterinary practice. The general objective of the research was explained to all owners, who signed a written, informed consent and agreed to have their animal included in the study and to the publication of the results. Participation was voluntary. In addition, they were informed that the data were collected and held anonymously and that confidentiality would be maintained by the researchers. Only data regarding the health and the behavior of the dogs were collected.

### 2.2. Data Collection

Data were collected from the specialist who performed the physiatric examination of the dogs and from the dog owners as follows:-*Demographic data:* Demographic data of the dogs were collected following the physiatric examination, together with the medical history.-*Clinical assessment:* In order to aid further analysis, the medical problem of the dog was classified as orthopedic, neurological, or degenerative (none of the dogs presented multiple diagnoses). The duration and type of physiotherapy treatment required by the clinician (hydrotherapy and/or instrumental techniques, such as laser therapy, massage, diathermic therapy) were also registered. The symptoms of the dog were recorded through a checklist measuring variation in body condition score (BCS), appetite, playfulness, interactions with the owner, and locomotion, as defined by the specialist. Each parameter was given a score of 1 (improved from before), 2 (same as before), or 3 (worsened). In addition, the clinician recorded lameness, pain levels, and overall criticality (1 = none, 2 = low, 3 = moderate, 4 = severe; Appendix A).-*Quality of life assessment*: As part of the clinical exam, the owners of the dogs were asked to complete the MPQL [14] at T_0_ (during the first consultation) and at T_1_ (at the end of the physiotherapy treatment) in order to monitor the welfare of the dogs. A higher score in the questionnaire indicated better QOL. Norms for each time are reported in the Appendix A.

### 2.3. Data Analysis

Analyses were performed using R statistical software [15]. The packages ordinal [16], rcompanion [17], and emmeans [18] were used.

The scores of the MPQL were calculated according to the literature [14], obtaining four domains (Physical QOL, Psychological QOL, Social QOL, and Environmental QOL), with a higher score indicating a better QOL in each domain. Some variables where owners could choose more than one option in the MPQL were categorized into signs of positive and negative affect, as indicated in the literature [14]. The predictive effect of demographic and clinical data, as well as the time (T_0_–T_1_), was assessed through a series of regressions. The main factor, time (T_0_–T_1_), was included in all models as the main comparison of interest. For the remaining predictors, before calculating the regression models, inferential tests were used to reduce model complexity. For each of the four QOL domains, the Kruskal–Wallis test was used with categorical data, the Mann–Whitney U test was used with binomial data, and the Spearman rho regression was used with continuous data in order to identify a relationship between the given variable and the domain. Ordinal logistic regression (OLR) models were then developed to assess the association between the QOL physical, psychological, social, and environmental scores (outcome variables) and each of the potential predictors for which there were significant differences in the preliminary analyses. A random factor (dog identity) was included in the models to account for repeated measures. Model fitting was tested using a likelihood ratio test and measured using the pseudo R^2^ calculated with the Nagelkerke method. The odds ratio for the predictors was calculated to evaluate the strength of such relationships. A two-sided *p* < 0.05 was considered as statistically significant.

## 3. Results

The model calculated for the outcome variable physical QOL explained 45% of the variance for this domain (Model fit: Nagelkerke R_p_^2^ = 0.35, AIC = 236.93, χ^2^(3) = 10.75, *p* = 0.001). A main effect without interaction was observed for the fixed factors “criticality” (1–3), “pain” (1–3), and “time” (T_0_ vs. T_1_). Specifically, each increasing point in criticality decreased the physical QOL score by a factor of 2.81 (mean score = 61.65, *p* = 0.009, Figure 1). Conversely, there were no significant differences associated with time (mean score: T_0_ = 60.25, T_1_ = 63.05) and pain level (mean score: no pain = 62.25, low = 66.00, moderate = 61.57, severe = 53.33; Table 1, Figure 1).

The model calculated for the outcome variable psychological QOL explained 49% of the variance for this domain (model fit: Nagelkerke R_p_^2^ = 0.49, AIC = 283.82, χ^2^(2) = 26.58, *p* < 0.001). A main effect was observed only for the factors “time” (T_0_ vs. T_1_, Figure 1) and “age”, with psychological QOL improving from T_0_ to T_1_ by a factor of 4.47 (mean score: T_0_ = 118.30, T_1_ = 127.60, *p* < 0.001) and age predicting a decline of 0.02 points in the psychological QOL score for every year of age of the dog (mean score: 122.95, *p* < 0.001, Table 1, Figure 1).

The model calculated for the outcome variable social QOL explained 28% of the variance for this domain (model fit: Nagelkerke R_p_^2^ = 0.28, AIC = 123.85, χ_4_ = 12.54, *p* = 0.014). The factors “medical problem” (orthopedic, neurological, degenerative) and “severity of lameness” (1–4) were included in the model. Post hoc analysis indicated that each increasing point in the severity of lameness decreased the psychological QOL score by a factor of 1.47 (mean score = 10.65, *p* = 0.016, Figure 1). Pairwise comparison indicated no significant differences in the social QOL measured in dogs suffering from each type of medical problem or with the time (Table 1).

In the case of the environmental QOL, no other factors were included in the model, which did not differ from a null model (model fit: Nagelkerke R_p_^2^ = 0.04, AIC = 219.03, χ_1_ = 1.64, *p* = 0.199), and there was no difference between T_0_ and T_1_ (mean score: T_0_ = 35.95, T_1_ = 36.80, Table 1, Figure 1).

None of the symptoms associated with the medical problems yielded significant differences in the dogs’ QOL (Appendix A Appendix A).

## 4. Discussion

The aim of this study was to investigate the changes in the QOL in dogs following a physiotherapy treatment, measured by a multi-domain QOL assessment tool, the MPQL. As we predicted, the psychological QOL of the dogs improved with the treatment. This goes to support the recent indications that physical pain plays an important role in the emotional welfare of companion animals, which is manifested through their behavior, including the expression of problematic behavior, e.g., signs of fear or aggressive behavior [11,19]. This is a relatively new, albeit not unexpected, finding, which highlights once again the close relationship between psychological and physical well-being [19,20,21]. Recent literature indicates that, in Italy, where the current study took place, up to 80% of cases seen by behavioral medicine specialists may have pain as the root cause for the behavior problem [19]. In addition, a recent qualitative study has given rise to concerns about a potential link between pain and certain cases of noise sensitivities. Signs such as a generalization of fear to several stimuli and environments and avoidance of other dogs are overrepresented in dogs affected by musculoskeletal pain, as opposed to dogs without pain [21]. In fact, the authors suggest that all dogs presenting with noise sensitivity should be screened for chronic pain. Similarly, pain is associated with certain aggressive incidents in dogs: dogs in pain are described as having a bad temperament (“Jekyll and Hyde personality”). Aggressive episodes for these dogs may have various triggers, although often the dog reacts when it is laying down and is approached by others, due to the fear of pain. Episodes are usually short and easy to break up as the goal of the dog is to prevent and/or interrupt the potentially painful interaction [20]. Dog owners also recognize the impact of chronic conditions, such as chronic inflammatory enteropathy, on quality of life [10]. Another important example of how painful conditions can be the cause of negative affect in dogs was provided by a study of mood in dogs affected by syringomyelia, a condition where a cavity forms in the spinal cord of the dog. The study used a cognitive test, called a cognitive bias test, to show that dogs affected by syringomyelia have a bias towards negative affect, likely due to the constant pain that they endure [22]. The cognitive bias test is a cognitive test that measures the bias that an individual has towards ambiguous stimuli. A negative bias, i.e., the individual interprets ambiguous stimuli as potentially unpleasant is known to be a reliable indicator of a negative mood and, vice versa, a positive bias indicates a positive mood [23]. Mood is a persistent emotional state, which lasts days or weeks [24]. Positive and negative moods are considered important welfare aspects. Historically, animal welfare research has focused on the prevention of poor welfare; however, in the past decade, animal welfare research has highlighted the importance of ensuring positive welfare as well [23]. This is particularly important for the most vulnerable categories, such as animals with chronic health issues and elderly animals [9]. In fact, in the current study, as well as in recent research, we observed a relationship between the age of the dog and psychological QOL [14]. Previous research revealed that it may be difficult to use cognitive testing to measure affect, for example, in elderly dogs, as these tests require learning, which may be impaired with age [25]. The results of the current study indicate that the findings of the physical exam reflect the anamnestic observations that owners can provide through structured questions. The owners can see subtle changes in the behavior of the dogs, such as irritability, withdrawal, avoidance of certain surfaces or locations, and so on, which is difficult to observe during the consultation. In these aspects, the information provided by the owner is particularly important to obtain a broader picture of the effects of physical pain on the overall QOL of the dog. It was suggested in the past that dogs with certain behavior problems, such as noise sensitivity, should always be screened for physical pain [21]. Similarly, we argue that dogs with painful conditions should also be assessed for their behavioral and psychological well-being.

Contrary to our expectations, there was no significant difference in the scores of the QOL questionnaire for the physical, social, and environmental domains between the beginning and the end of the physiotherapy treatment. One possible explanation for this finding is that, within each domain, only a few aspects are affected by the presence of pain and reduced mobility of the dog. For example, the physical QOL, in addition to questions regarding pain and mobility, included parameters such hearing, vision, and reproductive health, which are not likely to be affected by the types of diagnoses included in the current study. The current results indicate a relationship between the criticality of the medical condition and physical QOL, suggesting that the questionnaire does measure to some extent the effects of the medical condition. Furthermore, the literature suggests that sensory decline can be recognized by dog owners [26] and that it is often associated with increased behavior problems in owners’ reports, especially for mature and older dogs [27]. Therefore, these are important parameters to be considered when evaluating QOL, which reflects the multidimensionality of the individual’s subjective experience. Similarly, the social QOL domain covers activities that might be affected by pain and lameness, such as play, but it also has questions about the amount of time spent with the pet or the training style. These are important aspects, which strongly affect dogs’ welfare [28,29,30,31]. Lameness did affect, overall, the social QOL domain in the current study. However, the time spent with the dog is probably related more to the lifestyle of the owner [14], while the training or educational style is likely strongly connected with the attitude and the relationship with the pet [30]. Finally, the environmental QOL domain included questions about the freedom and safety within the environment, as well as opportunities for learning. For example, owners were asked about the type of restraint that was used during walks (e.g., collar, harness, as well as fixed lead or extensible lead). These would certainly be affected by the requirements and restraints associated with orthopedic and neurological conditions. Nevertheless, the environmental QOL domain also records information about access in the house when left alone, safe restraint and shelter options in the garden, and training. All these aspects are likely affected by cultural differences, legal requirements, and attitudes towards dogs; for example, in the countryside, there are several independent houses with gardens, while, in urban areas, there is a large prevalence of buildings with no outdoor access [32]. In general, the legal requirements about dog keeping, leashing, or fencing of gardens are usually between countries [33].

Overall, the results support the recent evidence that physical health and pain have a role on the psychological well-being of the dog. In addition, the results highlight the importance of investigating psychological and emotional aspects of dogs’ QOL when treating orthopedic and neurological cases with physiotherapy.

## Figures and Tables

**Figure 1 vetsci-09-00335-f001:**
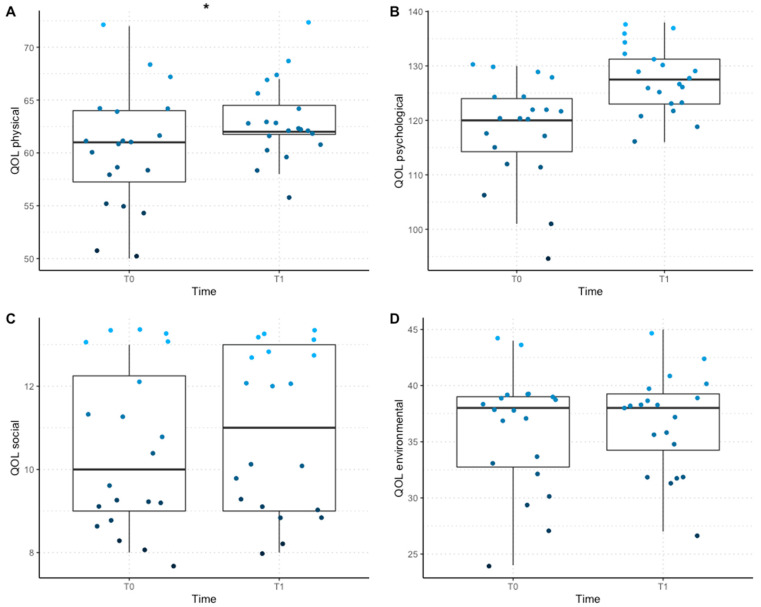
**Distribution of the QOL score for each domain, divided by responding time (T_0_ vs. T_1_).** The middle line in the box plots represents the median score, the extremes of the boxes represent the lower and upper quartiles, and the error bars represent the minimum and the maximum scores; * = *p* < 0.001. (**A**) Scores for the physical domain of QOL; (**B**) scores for the psychological domain of QOL; (**C**) scores for the social domain of QOL; (**D**) scores for the environmental domain of QOL.

**Table 1 vetsci-09-00335-t001:** Results of regression analyses on the MPQL. Note: * = *p* < 0.05.

	Estimate	S.E.	*p*
**CLMM Model Physical QoL**			
Time:			
T_0_ vs.T1	0.70	(0.95)	0.462
Pain:	−0.12	(0.51)	0.810
Criticality:	−2.82	(1.08)	0.009 *
**CLMM Model Psychological QoL**			
Time:			
T_0_ vs. T_1_	−4.47	(1.00)	<0.001 *
Age:	−0.02	(0.01)	<0.001 *
**CLMM Model Social QoL**			
Time:			
T_0_ vs. T_1_	0.25	(1.05)	0.808
Lameness:	−1.47	(0.61)	0.016 *
Medical problem:			
Degenerative vs. Neurological	−2.79	(3.28)	0.672
Degenerative vs. Orthopedic	−5.08	(2.97)	0.200
Neurological vs. Orthopedic	−2.29	(1.95)	0.467
**CLMM Model Environmental QoL**			
Time:			
T_0_ vs. T_1_	−0.80	(0.63)	0.204

## Data Availability

Data will be made available upon request to the authors.

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
