# Peer review of "Physiotherapy Improves Dogs’ Quality of Life Measured with the Milan Pet Quality of Life Scale: Is Pain Involved?"

_vetsci, 2022, doi:10.3390/vetsci9070335_

Round 1

Reviewer 1 Report

The interesting and essential topic concerning animal welfare.

The article requires changes to make it more readable.

I propose to edit the title and the Introduction section so that it is closely related to the purpose of the work.

Research methodology.

I suggest you check this section carefully.

Lines 98-117. Please specify the research methodology in the data analysis part. It is not known how the data was collected.

Discussion

I suggest authors study this section carefully. There are fragments that should be included in the Materials and Methods section (e.g. lines 206-209).

Lines 256-257. The sentence is not clear. Please explain: "legal requirements and attitudes towards dogs" fragment in particular.

Author Response

REVIEWER 1

The interesting and essential topic concerning animal welfare.

We would like to thank the reviewer for the appreciation.

The article requires changes to make it more readable.

I propose to edit the title and the Introduction section so that it is closely related to the purpose of the work.

We have modified the title adding the name of the scale that we used, and we included more details about the scale and about physiotherapy in the introduction (lines 64-67 and 84-86).

Research methodology.

I suggest you check this section carefully.

Lines 98-117. Please specify the research methodology in the data analysis part. It is not known how the data was collected.

We have added details about the data collection in the "data collection" and the "data analysis" paragraphs.

Discussion

I suggest authors study this section carefully. There are fragments that should be included in the Materials and Methods section (e.g. lines 206-209).

We thank the reviewer for the suggestion. However, we did not use the cognitive bias test in the current study, as it is discussed as a reference to previous literature to aid the interpretation of results.

Lines 256-257. The sentence is not clear. Please explain: "legal requirements and attitudes towards dogs" fragment in particular.

We have added further details to explain the concept (lines 279-282).

Reviewer 2 Report

This research shows recent evidence of a relationship between pain and canine psychological well-being and highlight the importance of investigating psychological and emotional aspects of dogs’ QOL when treating orthopaedic and neurological cases with physiotherapy. This study provides a lot of information to veterinarians and caregivers as it is an excellent study that suggests a new interpretation on postoperative pain reduction and quality of life in dogs. English of this paepr is also well described and will be accepted as is.

Author Response

Reviewer 2

We thank the reviewer for their words and we have worked to improve the manuscript.

Reviewer 3 Report

The authors attempted to explore the quality of life of the pet based on the data from two time points. This study is an interesting research topic. However, there are several issues need to be addressed by the authors:

1) The major finding of this study is a bit confused. In the abstract, the authors reported that “the study indicate a significant improvement in the psychological QOL domain following physiotherapic treatment… The results of the study support the recent evidence of a relationship between pain and canine psychological well-being” (p. 1). But then when the authors conceptualizing the MPQL on p. 2, pain is not included in any of the dimensions.

2) In the research design, many recent studies used control group or even RCT (Marchetti, Gori, Mariotti, Gazzano, & Mariti, 2021), please provide the justifications of not using this methods in the current study?

3) Please provide the sources to demonstrate the MPQL has been validated in the Italian context.

4) In the data analysis section (p. 3), please provide the justifications for using those procedure/equations for the longitudinal data (T0 and T1) (McIver, Hall, & Mills, 2020).

5) The variables used on Table 1, such as medical problem, age, pain, lameness need further conceptualization and operationalization in the methods section.

6) The hypothesis is also not clear, please further spell out the novelty and contribution of this study.

References

Marchetti, V., Gori, E., Mariotti, V., Gazzano, A., & Mariti, C. (2021). The Impact of Chronic Inflammatory Enteropathy on Dogs' Quality of Life and Dog-Owner Relationship. Veterinary Sciences, 8(8). doi:10.3390/vetsci8080166

McIver, S., Hall, S., & Mills, D. S. (2020). The Impact of Owning a Guide Dog on Owners’ Quality of Life: A Longitudinal Study. Anthrozoös, 33(1), 103-117. doi:10.1080/08927936.2020.1694315

Author Response

Reviewer 3

The authors attempted to explore the quality of life of the pet based on the data from two time points. This study is an interesting research topic. However, there are several issues need to be addressed by the authors:

We thank the reviewer and we have worked to improve the manuscript.

1) The major finding of this study is a bit confused. In the abstract, the authors reported that “the study indicate a significant improvement in the psychological QOL domain following physiotherapic treatment… The results of the study support the recent evidence of a relationship between pain and canine psychological well-being” (p. 1). But then when the authors conceptualizing the MPQL on p. 2, pain is not included in any of the dimensions.

We have specified in the introduction that the domain 1 (Physical QOL) includes questions of mobility changes that are considered due to pain (line 79).

2) In the research design, many recent studies used control group or even RCT (Marchetti, Gori, Mariotti, Gazzano, & Mariti, 2021), please provide the justifications of not using this methods in the current study?

We thank the reviewer for the consideration. We could not include a placebo group in the current study due to ethical reasons, because we involved actual patients in need for physiotherapy. Conversely, in Marchetti et al., 2021, which we have now cited in the text (lines 60 and 216), the authors looked at the impact of a disease, therefore it was possible to look at a group free from that condition. We agree that further research could look at control groups in ethical ways, such as comparing other treatments.

3) Please provide the sources to demonstrate the MPQL has been validated in the Italian context.

We have added this in the introduction (lines 84-86)

4) In the data analysis section (p. 3), please provide the justifications for using those procedure/equations for the longitudinal data (T0 and T1) (McIver, Hall, & Mills, 2020).

We thank the reviewer for the suggestion. We have used a similar analysis, but ordinal in nature as we were using ordinal data, as in the McIver et al 2020 paper.

5) The variables used on Table 1, such as medical problem, age, pain, lameness need further conceptualization and operationalization in the methods section.

We thank the reviewer for this consideration. The variables are explained in the clinical assessment section (lines 115-125) and each variable is described in details in the supplemental information (table S2).

6) The hypothesis is also not clear, please further spell out the novelty and contribution of this study.

We thank the reviewer for the suggestion, we have added this at lines 68-72.

References

Marchetti, V., Gori, E., Mariotti, V., Gazzano, A., & Mariti, C. (2021). The Impact of Chronic Inflammatory Enteropathy on Dogs' Quality of Life and Dog-Owner Relationship. Veterinary Sciences, 8(8). doi:10.3390/vetsci8080166

McIver, S., Hall, S., & Mills, D. S. (2020). The Impact of Owning a Guide Dog on Owners’ Quality of Life: A Longitudinal Study. Anthrozoös, 33(1), 103-117. doi:10.1080/08927936.2020.1694315

Round 2

Reviewer 1 Report

I thank the authors for the changes made.

Generally, the work has been revised as suggested by the reviewer.

According to the reviewer, the statistical analysis of the results is discussed in a very synthetic (general) way. It could be made more specific.

However, I leave the final decision to the Editor and accept the work without comments.